# STRUCTURED VIDEO-LANGUAGE MODELING WITH TEMPORAL GROUPING AND SPATIAL GROUNDING

**Yuanhao Xiong**[1,3*]   **Long Zhao**[1]   **Boqing Gong**[1]   **Ming-Hsuan Yang**[1]
**Florian Schroff**[1]   **Ting Liu**[1]   **Cho-Jui Hsieh**[2,3]   **Liangzhe Yuan**[1]
[1]Google Research   [2]Google   [3]UCLA

## ABSTRACT

Existing video-language pre-training methods primarily focus on instance-level alignment between video clips and captions via global contrastive learning but neglect rich fine-grained local information in both videos and text, which is of importance to downstream tasks requiring temporal localization and semantic reasoning. A powerful model is expected to be capable of capturing region-object correspondences and recognizing scene changes in a video clip, reflecting spatial and temporal granularity, respectively. To strengthen model's understanding into such fine-grained details, we propose a simple yet effective video-language modeling framework, S-ViLM, by exploiting the intrinsic structures of these two modalities. It includes two novel designs, inter-clip spatial grounding and intra-clip temporal grouping, to promote learning region-object alignment and temporal-aware features, simultaneously. Comprehensive evaluations demonstrate that S-ViLM performs favorably against existing approaches in learning more expressive representations. Specifically, S-ViLM surpasses the state-of-the-art methods substantially on four representative downstream tasks, covering text-video retrieval, video question answering, video action recognition, and temporal action localization.

## 1  INTRODUCTION

Videos are composed of groups of pixels spanning spatially and temporally. Semantically related groups of pixels form the objects in the visual scenes and their changes through space and time vividly show the action and interactions of physical world. Scene switching further complicates the video story line and finally depicts attractive video stories. The similar structures also appear in the paragraphs when they come to describe the videos. Captions are built from the basic grammar components such as nouns and verbs, and sentences are concatenated to describe complex scenes.

Modern video-language models (VLMs), however, mostly neglect the fine-grained structures of such video-text pairs during the development. Video-language pre-training typically follows the pipeline: (1) encoding video and text pairs into latent representations, (2) modality fusion, and (3) pre-training on specific objectives. Existing methods typically optimize these three components in the pre-training pipeline by designing expressive encoders (Bain et al., 2021; Li et al., 2022; Ge et al., 2022a; Nagrani et al., 2022; Ma et al., 2023), fusing two modalities via a cross-encoder (Ge et al., 2022a; Li et al., 2022; Lei et al., 2021; Li et al., 2020; Luo et al., 2020; Xu et al., 2021a; Zhu & Yang, 2020), or adopting a combination of various pre-training tasks such as contrastive learning and masked modeling (Li et al., 2022; Ge et al., 2022a; Fu et al., 2021; Zellers et al., 2022; Cao et al., 2022; Ge et al., 2022b). While these modifications benefit the pre-trained model, their lack of local discriminative modeling poses challenges for VLMs to further understand complex videos.

It has been shown that most video-language pre-training methods merely perform well on learning holistic representations to match a ⟨*video, caption*⟩ pair while neglect fine-grained information such as region-object correspondences, or scene/action changes along the time in a video (Akbari et al., 2021; Bain et al., 2021; Lei et al., 2021; Li et al., 2020; Luo et al., 2020; Miech et al., 2019; Xu et al., 2021b; Nagrani et al., 2022). However, such regional or temporal fine-grained information has been demonstrated to play a vital role in localization and reasoning tasks (Li et al., 2022; Ge et al., 2022a;

---

*Work done as a student researcher at Google Research.

Zhang et al., 2022; Ma et al., 2023; Yuan et al., 2022). Motivated by aforementioned observations, we revive the strong connectivity between basic components of video clips and languages during self-supervised video-language pre-training. We approach the video-language pre-training task from a different perspective with a focus on exploiting spatiotemporally fine-grained structures.

In this work, we incorporate structured video-language interactions into the pre-training stage and propose a novel framework, namely **S**tructured **Vi**deo-**L**anguage **M**odeling (S-ViLM), with temporal grouping and spatial grounding. S-ViLM encourages instance-level video-caption alignment, fine-grained region-object alignment, and learns temporal-aware video representations, simultaneously. As shown in Figure 1, S-ViLM consists of three primary training objectives: inter-clip spatial grounding, intra-clip temporal grouping, and global contrastive learning. Given a video-caption pair as the input, a classical dual-encoder model is leveraged to extract the representation for each modality, respectively. Videos are pre-processed with the cut-and-paste operation, inspired by (Zhang et al., 2022; Yun et al., 2019), i.e., pasting one clip in a video onto the other background video, to explicitly introduce temporal scene changes. We further adopt grouping blocks (Xu et al., 2022; Yu et al., 2022) to aggregate semantically similar video patches to represent regions without off-the-shelf detectors via a set of group tokens shared among all videos. In *inter-clip spatial grounding*, we align grouped video tokens with objects represented by nouns in the caption by minimizing our designed grounding loss. In *intra-clip temporal grouping*, we improve features temporal granularity by distinguishing foreground and background representations within one clip. Finally, the model is trained by a global video-caption contrastive loss to match instance-level video-caption pairs. We evaluate our proposed method comprehensively on four representative tasks, including text-video retrieval, video question answering, video action recognition, and temporal action localization. Our strong experimental results demonstrate that exploiting fine-grained video-text structures during pre-training effectively improves VLM's video understanding and reasoning capabilities.

Our key contributions are summarized as follows:

- We propose S-ViLM, a dual-encoder video-language modeling framework, making use of structured video-caption interactions to learn more expressive spatiotemporal features.

- We leverage a cut-and-paste operation to introduce scene changes into videos during pre-training, and propose an intra-clip grouping module to learn more temporal-aware features.

- We design an inter-clip spatial grounding module to capture fine-grained correspondences by aligning objects from the caption and regions from the video in a self-supervised manner.

- Experimental results have demonstrated the effectiveness of S-ViLM on four downstream tasks, including text-video retrieval, video question answering, video action recognition, and temporal action localization. For example, S-ViLM outperforms SOTA by 3% in R@1 in zero-shot video-text retrieval on MSR-VTT and 5% in accuracy in action recognition on UCF101, showing its advantages over both multi-modal and single-modal tasks.

## 2  RELATED WORK

**Video-language pre-training.** Video-language pre-training is an emerging research area that aims to develop machine learning models capable of jointly understanding visual and textual content. Representations learned from large scale noisy datasets such as HowTo100M (Miech et al., 2019), WebVid (Bain et al., 2021), and VideoCC (Nagrani et al., 2022) have demonstrated great potentials in adapting to downstream tasks, including but not limited to text-video retrieval, video question answering, and video captioning. Elaborately designed pre-training objectives ranging from generative (Chen et al., 2020b; Fu et al., 2021; Li et al., 2019; Liu et al., 2022) to discriminative (Bain et al., 2021; Lei et al., 2021; Akbari et al., 2021; Sun et al., 2022; Li et al., 2022; Ge et al., 2022a; Ma et al., 2022; Wang et al., 2023a;b;c) have been proposed, among which contrastive learning is prevalent and widely adopted to attract paired video-caption instances and repelling unpaired ones. However, their primary focus is still on learning holistic global representations to align instance-level ⟨*video, caption*⟩ pairs. Recently, some approaches have been proposed to leverage finer-grained information such as nouns/verb phrases from a caption. ALPRO (Li et al., 2022) extracts pseudo entity labels by feeding noun prompts into a frozen model and use contrastive objective to align cropped visual regions and the corresponding textual labels. In Ge et al. (2022a), MCQ recovers randomly masked noun/verb tokens via resorting to global video features, which implicitly improves text entity

association in visual encoding. LAVILA (Zhao et al., 2023) constructed temporally dense captions by automatic annotation from large language models to describe activities more comprehensively. In addition, TemPVL (Ma et al., 2023) enables temporal and semantic alignment such that the trained model can accurately perceive temporal boundaries in videos given the text description. Despite these efforts, correspondences between visual regions and objects from noun concepts in captions and temporal scene shifts in a video are still neglected and not modeled explicitly in existing video-language pre-training methods. In this work, we propose two novel designs, spatial grounding and temporal grouping, to leverage fine-grained information in the pre-training stage.

**Vision language grounding.** The goal of visual grounding (VG) is to locate the most relevant object or region in a visual input based on a natural language query (Fang et al., 2015; Rohrbach et al., 2016; Fukui et al., 2016; Ghiasi et al., 2022; Gupta et al., 2020). Recently, visual grounding has been adapted to pre-training tasks in a self-supervised manner for open-vocabulary image segmentation (Ghiasi et al., 2022; Xu et al., 2022). For example, OpenSeg (Ghiasi et al., 2022) semantically aligns a caption with extracted image regions via a grounding loss. Moreover, without the off-the-shelf object detectors, GroupViT (Xu et al., 2022) learns to group together semantic regions from text supervision by contrastive learning. Note that visual grounding is mostly discussed in the image domain and its success motivates us to extend visual-semantic alignment to video-language pre-training. To achieve this, we integrate a novel spatial grounding module in our framework to promote visual and textual entity correspondences in a self-supervised manner.

**Video temporal modeling.** In contrast to images, videos contain a sequence of dynamic frames and how to model temporal information is critical in video understanding (Feichtenhofer et al., 2019; Bertasius et al., 2021; Tran et al., 2014; Alwassel et al., 2021; Zhang et al., 2022; Qian et al., 2022). Specifically, TSP (Alwassel et al., 2021) learns temporal information via predicting clips inside or outside the action with substantial annotations. PAL (Zhang et al., 2022) aligns features of pasted pseudo action regions from two synthetic videos. BSP (Xu et al., 2021c) introduces a novel boundary-sensitive pretext task via classifying the boundary types of synthetic videos. These techniques are elaborately designed for training models on long videos such as movies or TV dramas, which contains natural scene changes. However, few of them have been considered in video-language pre-training since the majority of video-language datasets contains short videos with repeated frames and are lacking in temporal differences. Instead, we develop a temporal grouping method to learn temporal-aware clip features in a self-supervised manner. We show that features extracted from explicitly temporal modeling achieve significant improvements in not only temporal action localization tasks, but also coarse-grained reasoning and understanding tasks such as video question answering and video action recognition.

## 3 METHOD

### 3.1 OVERVIEW

The framework of S-ViLM is presented in Figure 1. We adopt the dual encoder architecture for video-language pre-training, and there are three primary objectives used in the pre-training stage: (1) inter-clip spatial grounding, (2) intra-clip temporal grouping, and (3) global contrastive learning.

As shown in Figure 1, temporal changes are first artificially introduced into training examples through cut-and-paste. Then the pre-processed video together with learnable group tokens are fed into the video encoder. Specifically, group tokens aggregate semantically similar video tokens via grouping blocks and are then aligned with object concepts by spatial grounding. It promotes region-object groundingness, which indicates the alignment between a region in the video and an object in the caption, e.g., as illustrated in Inter-clip Spatial Grounding in Figure 1, the red region corresponds exactly to the word "pins" in red. In contrast to previous methods where regions are extracted with pre-trained object detectors (Cai et al., 2022; Li et al., 2022; Yan et al., 2021), these learnable group tokens can cluster and organize semantically similar regions in a self-supervised manner, which is more effective and reduces the artifacts of any detectors. For the language branch, the original captions are tokenized into a sequence of text tokens, which are then fed into a text encoder to extract the corresponding representation from the preceding `[CLS]` token. Noun tokens representing objects are extracted in the same way given a set of prompts.

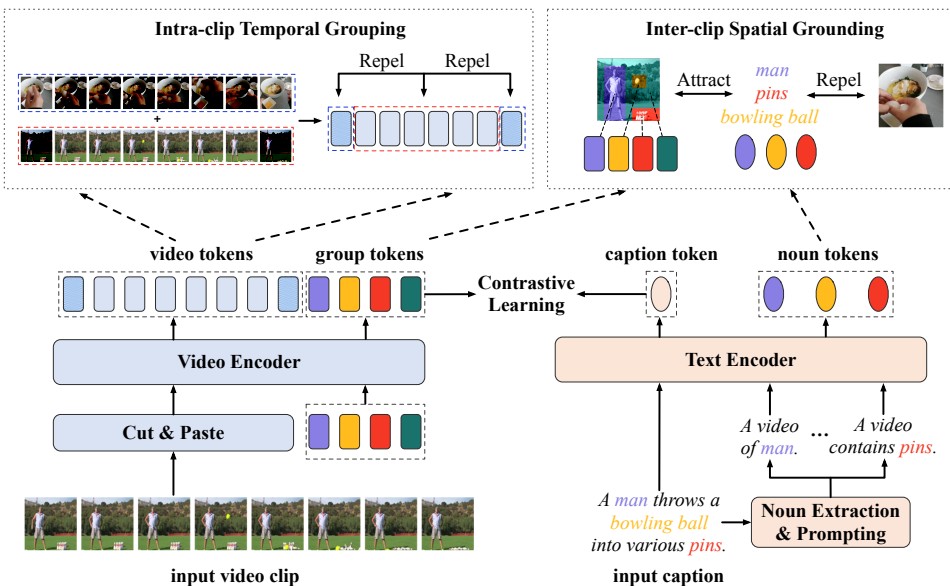

Figure 1: Illustration of S-ViLM pre-training. Three proposed training objectives promote structured video-language interaction: (1) temporal grouping learns temporal-aware features by distinguishing whether clips are from background or foreground; (2) spatial grounding focuses on local correspondences between regions and objects; (3) global contrastive learning matches instance-level ⟨*video, caption*⟩ pairs.

To promote temporal awareness, we use masks derived from the cut-and-paste operations as the ground-truth for temporal grouping. Furthermore, we model the interaction between region features and noun tokens using inter-clip spatial grounding loss. Finally, a global contrastive loss is computed between the video and the caption representations to match the instance-level ⟨*video, caption*⟩ pair.

## 3.2 INTRA-CLIP TEMPORAL GROUPING WITH CUT-AND-PASTE

Commonly-used video-language pre-training data usually consist of short video clips with repetitive scenes. To simulate scene shifts, we design a cut-and-paste operation inspired from image augmentations (Yun et al., 2019; Zhang et al., 2022) to introduce temporal changes manually to further improve video representations.

Given a target video $v_i$ with $T$ frames as the foreground and a randomly sampled video $v_{p_i}$ with the index $p_i$ as the background from the same batch of size $B$, we divide each video into $N_t = T/t$ clips with the temporal window size $t$. We then sample the start and end clip indices $s$ and $e$ from $(0, N_t)$, and paste the corresponding region from $v_i$ into the background video $v_{p_i}$ to form a blended video $\hat{v}_i$. For the clip sampling procedure, we first uniformly sample the duration of the foreground video $d$ from $[N_t/2, N_t)$ to guarantee it is the majority of the blended video. Then we sample the start index $s$ from $[0, N_t - d)$, and the end index $e$ was computed naturally as $e = s + d$. We included this detail in our latest version. We define the foreground-background mask as $m_i \in \mathbb{R}^{N_t} = \{\mathbf{1}(j \in [s, e]) | j \in [0, N_t)\}$, where $\mathbf{1}(\cdot)$ is the indicator function. This operation is illustrated in Figure 1.

A video is first flattened into $N$ non-overlapping voxels. After projected by a linear layer, these voxel tokens are fed into the transformer encoder to obtain transformed tokens $z_i^v \in \mathbb{R}^{N \times d}$, where $d$ is the feature dimension. To obtain clip-level representations $z_i^{\text{clip}} \in R^{N_t \times d}$, we average-pool over $z_i^v$ along the spatial dimension after recovering the feature map's 3D shape. Two cluster centers, $z_i^b$ for the background and $z_i^f$ for the foreground, are further computed by averaging features from $z_i^v$ on the corresponding position based on the mask $m_i$. To assign each clip to either background or foreground, we compute $a_i$ via cosine similarity with an element-wise softmax function applied on the last dimension, where $\langle \cdot, \cdot \rangle$ is cosine similarity and $\tau$ is the temperature to scale logits:

$$a_i = \text{Softmax}(\langle z_i^{\text{clip}}, [z_i^b; z_i^f]^T \rangle / \tau) \in \mathbb{R}^{N_t \times 2}. \tag{1}$$

Finally, the temporal grouping loss can be computed within a batch between $a_i$ and the ground-truth one-hot masking $m_i$ using mean squared error as

$$\mathcal{L}_t = \frac{1}{B} \sum_i^B \ell_{\text{BCE}}(a_i, \text{One-hot}(m_i)). \tag{2}$$

Note that we have also tried the binary cross entropy loss which performs comparably to MSE. Thus, we select a relatively simple MSE loss for temporal grouping.

## 3.3 Inter-clip Spatial Grounding with Group Tokens

Observing the correspondences between visual regions in a video and noun phrases (objects) in a caption, we model such fine-grained alignment for more expressive encoders. In practice, it is infeasible to pool tokens of interest as cluster centers since we do not have ground-truth segmentation. Thus, we adopt $M$ learnable group tokens to cluster semantically similar regions in a self-supervised manner. Note that group tokens are randomly initialized and shared among different videos. The detailed structure of a grouping block is presented in Appendix A and multiple grouping blocks are placed at different layers of the video encoder to update group tokens progressively. The final group tokens denoted as $\mathcal{G} = \{g_i^m\}_{m=1}^M$ aggregate semantically similar voxels and represent different regions in the video $v_i$. Compared with using off-the-shelf region proposal networks, our design of token grouping is more computationally efficient, and can be adapted to the pre-training dataset without region annotations in a self-supervised manner dynamically and flexibly. For each caption $c_i$ of a video, we extract $K$ noun phrases using noun chunking in spaCy[1] and prompt each of them with a set of handcrafted sentence templates, e.g., "*A photo of a {noun}*". Such prompted noun phrases are fed into the text encoder to extract noun tokens $\{n_i^k\}_{k=1}^K$.

We define the notation for softmax on a vector $\mathbf{x}$ at the $i$-th element as: $\sigma(\mathbf{x})_i = \frac{\exp(x_i)/\tau}{\sum_j \exp(x_j)/\tau}$, where $\tau$ is the temperature to scale logits. The similarity of all group tokens $\mathcal{G}$ with respect to a noun token $n^k$ is defined as $s(\mathcal{G}, n^k) = [\langle g^1, n^k \rangle, \dots, \langle g^M, n^k \rangle] \in \mathbb{R}^M$, where $\langle \cdot, \cdot \rangle$ is the cosine similarity. Since the ground-truth correspondences between regions and nouns are inaccessible, we compute the grounding similarity between all group and noun tokens by:

$$G(v, c) = \frac{1}{K} \sum_{k=1}^K \left\langle n^k, \sum_{m=1}^M \sigma\left(s(\mathcal{G}, n^k)\right)_m \cdot g^m \right\rangle. \tag{3}$$

$G(v, c)$ encourages each noun to be grounded to one or a few regions and avoids penalizing regions that cannot find any relevant nouns.

Similarity scores over a batch of size $B$ are computed as: $G(\mathcal{V}, c_i) = [G(v_1, c_i), \dots, G(v_B, c_i)] \in \mathbb{R}^B$ and $G(v_i, \mathcal{C}) = [G(v_i, c_1), \dots, G(v_i, c_B)] \in \mathbb{R}^B$, where $\mathcal{V} = \{v_i\}_{i=1}^B$ and $\mathcal{C} = \{c_i\}_{i=1}^B$ denote the set of videos and captions in a batch, respectively. Inter-clip spatial grounding loss $\mathcal{L}_g$ is then defined to enable nouns to be matched with regions for each positive ⟨*video, caption*⟩ pair: $\mathcal{L}_g = \mathcal{L}_g^{v \to c} + \mathcal{L}_g^{c \to v}$ consists of a video-to-caption grounding loss and a caption-to-video grounding loss

$$\mathcal{L}_g^{v \to c} = -\frac{1}{B} \sum_{i=1}^B \log \sigma\left(G\left(v_i, \mathcal{C}\right)\right)_i, \quad \mathcal{L}_g^{c \to v} = -\frac{1}{B} \sum_{i=1}^B \log \sigma\left(G\left(\mathcal{V}, c_i\right)\right)_i. \tag{4}$$

Recall that the cut-and-paste operation indicates that $\hat{v}_i$ has another positive caption $c_{p_i}$ besides its original $c_i$, and the loss weights of positive indices are $W^v \in \mathbb{R}^{B \times B} = \{w_{i,j}^v\}$ which satisfy

$$w_{i,j}^v = \begin{cases} \beta_i, & j = i \\ 1 - \beta_i, & j = p_i \\ 0, & \text{otherwise} \end{cases}, \tag{5}$$

where $\beta_i = (e - s)/N_t$ is the ratio of the foreground in the cut-and-paste video $\hat{v}_i$. From the perspective of captions, we can obtain $W^c = (W^v)^\top$. We can derive the augmented grounding loss $\mathcal{L}_g$ with the video-to-caption loss and and the caption-to-video loss:

$$\mathcal{L}_g^{v \to c} = -\frac{1}{B} \sum_{i=1}^B \sum_{j=1}^B w_{i,j}^v \log \sigma\left(G(\hat{v}_i, \mathcal{C})\right)_j, \quad \mathcal{L}_g^{c \to v} = -\frac{1}{B} \sum_{i=1}^B \sum_{j=1}^B w_{i,j}^c \log \sigma\left(G\left(\hat{\mathcal{V}}, c_i\right)\right)_j. \tag{6}$$

[1] https://spacy.io/

## 3.4 Overall Pre-training Objective

We include a global contrastive learning objective for instance-level alignment. $f_i^v$, the video representation of $\hat{v}_i$, is extracted from average-pooled group tokens and $f_i^c$, the caption representation $c_i$, is computed from the [CLS] token of the original caption. Instance similarity scores are defined as: $s(\mathcal{V}, c_i) = [\langle f_1^v, f_i^c \rangle, \ldots, \langle f_B^v, f_i^c \rangle] \in \mathbb{R}^B$ and $s(\hat{v}_i, \mathcal{C}) = [\langle f_i^v, f_1^c \rangle, \ldots, \langle f_i^v, f_B^c \rangle] \in \mathbb{R}^B$. A global contrastive loss is defined as $\mathcal{L}_{\text{contrast}} = \mathcal{L}_{\text{contrast}}^{v \to c} + \mathcal{L}_{\text{contrast}}^{c \to v}$, a combination of the video-to-caption and the caption-to-video views:

$$\mathcal{L}_{\text{contrast}}^{v \to c} = -\tfrac{1}{B} \sum_{i=1}^{B} \sum_{j=1}^{B} w_{i,j}^v \log \sigma(s(\hat{v}_i, \mathcal{C}))_j, \quad \mathcal{L}_{\text{contrast}}^{c \to v} = -\tfrac{1}{B} \sum_{i=1}^{B} \sum_{j=1}^{B} w_{i,j}^c \log \sigma(s(\mathcal{V}, c_i))_j. \quad (7)$$

The overall pre-training objective is a weighted sum of grouping loss, grounding loss, and global contrastive loss: $\mathcal{L} = \omega_1 \mathcal{L}_t + \omega_2 \mathcal{L}_g + \omega_3 \mathcal{L}_{\text{contrast}}$. We set three weights, $\omega_1$, $\omega_2$, and $\omega_3$, to be equal to one in our experiments for simplicity.

## 4 Experiments

### 4.1 Downstream Tasks

**Text-video retrieval.** We adopt the widely used text-video retrieval benchmark MSR-VTT (Xu et al., 2016) for evaluation. It consists of 10K YouTube video clips with 200K captions. We conduct experiments in both zero-shot and fine-tuning settings. For fine-tuning setup, we follow Bain et al. (2021) and Ge et al. (2022a), and train and test the model on the split of 9K and 1K videos.

**Video question answering (VQA).** We consider open-ended VQA settings with two representative datasets: (1) MSRVTT-QA (Xu et al., 2017) with 1,500 answer candidates and (2) MSVD-QA (Xu et al., 2017) with 2,423 answer candidates.

**Video action recognition.** We select HMDB51 (Kuehne et al., 2011) containing 6,766 videos with 51 categories and UCF101 (Soomro et al., 2012) containing 13,320 videos with 101 categories. Both linear probing and fine-tuning the whole model are explored.

**Temporal action localization (TAL).** TAL aims at predicting the temporal extent and the labels of action instances. We evaluate the performance on ActivityNet (Heilbron et al., 2015), an action understanding dataset of 19,994 temporally annotated untrimmed videos with 200 action categories.

### 4.2 Implementation Details

**Input.** Following Nagrani et al. (2022), we sample 32 frames for each video and resize them into $224 \times 224$ with the same augmentations. Each caption is tokenized into 32 tokens including [CLS]. $K = 2$ noun phrases are extracted for each caption and then prompted with a set of prompt templates such as *"It is a video of {noun}"*. We include the full list of prompt templates in Appendix A.

**Model architecture.** We use a 12-layer ViT-base model with the patch size of $2 \times 16 \times 16$ as the video encoder and initialize it with weights pre-trained on Kinetics-400. We adopt 32 learnable group tokens and 3 grouping blocks featuring K-means attention (Xu et al., 2022; Yu et al., 2022). Grouping blocks are inserted at the 6th, 9th and last layers of the video encoder (Xu et al., 2022; Yu et al., 2022). The text encoder is initialized from the pre-trained BERT-base model. All representations are projected into the common space with the dimension of 256.

**Pre-training datasets.** We pre-train S-ViLM with the VideoCC (Nagrani et al., 2022) dataset, which contains about 3.3M video-caption pairs. We also include ActivityNet-Caption (Krishna et al., 2017) with 20K well-aligned pairs into the pre-training corpus. We note the commonly-used WebVid (Bain et al., 2021) is unavailable to us due to the restricted data access policy. To illustrate the effectiveness of our proposed method and how the pre-training datasets contribute to the final results, we designed fair studies on dataset impacts. Details could be found in Section 4.3.5.

**Pre-training and fine-tuning setups.** We implement S-ViLM in JAX and train all models on TPU accelerators. During pre-training, SGD with momentum 0.9 and initial learning rate 0.1 is used for optimization. We train S-ViLM for 10 epochs with a batch size 1024 and adopt a cosine learning rate decay schedule with a warmup ratio 0.05. It takes about one day for the whole pre-training stage. In

Table 1: Zero-shot (top) and fine-tuning evaluation (bottom) of text-video retrieval on MSR-VTT test set with 1K videos. **Higher** R@k and **lower** MedR (Median Rank) indicate better performance.

| Method | Video Encoder Input | PT Dataset | #Pairs PT | R@1 | R@5 | R@10 | MedR |
|---|---|---|---|---|---|---|---|
| MIL-NCE (Miech et al., 2019) | Raw Videos | HowTo100M | 120M | 9.9 | 24.0 | 32.4 | 29.6 |
| VATT (Akbari et al., 2021) | Raw Videos | HowTo100M, AudioSet | 138M | - | - | 29.7 | 49.0 |
| VideoCLIP (Xu et al., 2021b) | S3D | HowTo100M | 110M | 10.4 | 22.2 | 30.0 | - |
| SupportSet (Patrick et al., 2020) | R(2+1)D-34 | HowTo100M | 120M | 12.7 | 27.5 | 36.2 | 24.0 |
| Frozen (Bain et al., 2021) | Raw Videos | CC3M, WebVid-2M | 5.5M | 18.7 | 39.5 | 51.6 | 10.0 |
| AVLnet (Rouditchenko et al., 2021) | ResNeXt-101 | HowTo100M | 120M | 19.6 | 40.8 | 50.7 | 9.0 |
| DemoVLP (Cai et al., 2022) | Raw Videos | CC3M, WebVid-2M | 5.5M | 24.0 | 44.0 | 52.6 | 8.0 |
| ALPRO (Li et al., 2022) | Raw Videos | CC3M, WebVid-2M | 5.5M | 24.1 | 44.7 | 55.4 | 8.0 |
| MCQ (Ge et al., 2022a) | Raw Videos | CC3M, WebVid-2M | 5.5M | 26.0 | 46.4 | 56.4 | 7.0 |
| VCC (Nagrani et al., 2022) | Raw Videos | VideoCC | 3.3M | 18.9 | 37.5 | 47.1 | - |
| **S-ViLM** | Raw Videos | VideoCC, ActivityNet | 3.3M | **28.6** | **53.6** | **65.1** | **5.0** |
| UniVL (Luo et al., 2020) | S3D | HowTo100M | 110M | 21.2 | 49.6 | 63.1 | 6.0 |
| MMT (Gabeur et al., 2020) | S3D | HowTo100M | 120M | 26.6 | 57.1 | 69.6 | 4.0 |
| ClipBERT (Lei et al., 2021) | Raw Videos | COCO, VisGenome | 5.6M | 22.0 | 46.8 | 59.9 | 6.0 |
| AVLnet (Rouditchenko et al., 2021) | ResNeXt-101 | HowTo100M | 120M | 27.1 | 55.6 | 66.6 | 4.0 |
| SupportSet (Patrick et al., 2020) | R(2+1)D-34 | HowTo100M | 120M | 30.1 | 58.5 | 69.3 | 3.0 |
| VideoCLIP (Xu et al., 2021b) | S3D | HowTo100M | 110M | 30.9 | 55.4 | 66.8 | - |
| Frozen (Bain et al., 2021) | Raw Videos | CC3M, WebVid-2M | 5.5M | 31.0 | 59.5 | 70.5 | 3.0 |
| DemoVLP (Cai et al., 2022) | Raw Videos | CC3M, WebVid-2M | 5.5M | 36.0 | 61.0 | 71.8 | 3.0 |
| ALPRO (Li et al., 2022) | Raw Videos | CC3M, WebVid-2M | 5.5M | 33.9 | 60.7 | 73.2 | 3.0 |
| MCQ (Ge et al., 2022a) | Raw Videos | CC3M, WebVid-2M | 5.5M | 37.6 | 64.8 | 75.1 | 3.0 |
| VIOLETv2 (Fu et al., 2023) | Raw Videos | CC3M, WebVid-2M | 5.5M | 37.2 | 64.8 | 75.8 | - |
| All-in-One (Wang et al., 2023b) | Raw Videos | HowTo100M, WebVid-2M | 112M | 37.1 | **66.7** | 75.9 | - |
| VCC (Nagrani et al., 2022) | Raw Videos | VideoCC | 3.3M | 35.0 | 63.1 | 75.1 | - |
| **S-ViLM** | Raw Videos | VideoCC, ActivityNet | 3.3M | **38.4** | 65.7 | **76.3** | **2.0** |

terms of fine-tuning, different tasks are trained independently with their own set of hyperparameters on the target dataset and more details can be found in Appendix A. For temporal action localization, we fix weights of the pre-trained video encoder and its grouping blocks to extract video features, which are then evaluated by G-TAD (Xu et al., 2020), a commonly used method for TAL.

## 4.3 EVALUATION RESULTS

### 4.3.1 TEXT-VIDEO RETRIEVAL

We evaluate S-ViLM for the text-video retrieval task on MSR-VTT under both zero-shot and fine-tuning settings, and compare it with existing prevalent methods in Table 1. S-ViLM outperforms other methods significantly for zero-shot evaluation with R@10 of 65.1, yielding approximately 9% improvement over the best-performing baseline MCQ. The superior results demonstrate that our pre-trained model builds up a good alignment between video and language and generalizes well to unseen datasets. S-ViLM also achieves performance gain when the model is fine-tuned on the target MSR-VTT dataset, which further validates advantages of the pre-trained model. Note that S-ViLM performs favorably against existing methods despite the much smaller size of the pre-training data used in S-ViLM than those in baselines, such as HowTo100M and WebVid-2M.

### 4.3.2 VIDEO QUESTION ANSWERING

VQA results on two open-ended datasets are shown in Table 2. To enable S-ViLM to deal with the VQA task, we add a fusion head adapted from BUTD (Anderson et al., 2018) by integrating video and text features with simple linear layers. Then a classifier is inserted after the fusion module to perform question answering as a classification problem. Compared with previous methods which leverage particular architectures for VQA or include a complicated fusion encoder, S-ViLM is the most efficient and flexible for various vision-language tasks. S-ViLM achieves better performance than competing methods with the accuracy of 43.5% (+1.4%) and 46.4% (+0.5%) on MSRVTT-QA and MSVD-QA, respectively.

### 4.3.3 VIDEO ACTION RECOGNITION

For video action recognition, we only keep the video encoder together with its grouping blocks to extract single-modality video representations for evaluation. Two evaluation settings are considered: (1) linear probing where the backbone encoder is frozen and only the last linear classifier is trained and (2) end-to-end fine-tuning where both the backbone and the classifier are trained. Top-1 accu-

Table 2: Top-1 accuracy (%) of Video Question Answering on MSRVTT-QA and MSVD-QA.

| Method | PT Dataset | MSRVTT-QA | MSVD-QA |
|---|---|---|---|
| HGA (Jiang & Han, 2020) | - | 35.5 | 34.7 |
| QUEST (Jiang et al., 2020) | - | 34.6 | 36.1 |
| HCRN (Le et al., 2020) | - | 35.6 | 36.1 |
| ClipBERT (Lei et al., 2021) | COCO, VG | 37.4 | - |
| SSML (Amrani et al., 2021) | HowTo100M | 35.1 | 35.1 |
| CoMVT (Seo et al., 2021) | HowTo100M | 39.5 | 42.6 |
| DemoVLP (Cai et al., 2022) | CC3M, WebVid-2M | 38.3 | 39.5 |
| ALPRO (Li et al., 2022) | CC3M, WebVid-2M | 42.1 | 45.9 |
| **S-ViLM** | VideoCC, ActivityNet | **43.5** | **46.4** |

Table 3: Experiments of action recognition on UCF101 and HMDB51 with linear evaluation (Lin) and fully fine-tuning evaluation (FT).

| Method | Modal | UCF101 | | HMDB51 | |
|---|---|---|---|---|---|
| | | Lin | FT | Lin | FT |
| CoCLR (Han et al., 2020) | OF | 77.8 | 90.6 | 52.4 | 62.9 |
| MVCGC (Huo et al., 2021) | MV | 78.0 | 90.8 | 53.0 | 63.4 |
| XDC_R (Alwassel et al., 2020) | A | 80.7 | 88.8 | 49.9 | 61.2 |
| XDC_K (Alwassel et al., 2020) | A | 85.3 | 91.5 | 56.0 | 63.1 |
| MIL-NCE (Miech et al., 2019) | T | 83.4 | 89.1 | 54.8 | 59.2 |
| Frozen (Bain et al., 2021) | T | 87.8 | 89.8 | 61.3 | 66.3 |
| VATT (Akbari et al., 2021) | A, T | 89.2 | - | 63.3 | - |
| ELO (Piergiovanni et al., 2020) | A, OF | - | 93.8 | 64.5 | 67.4 |
| MMV (Alayrac et al., 2020) | A | 77.1 | - | 53.6 | - |
| MMV (Alayrac et al., 2020) | T | 86.8 | - | 55.1 | - |
| MMV (Alayrac et al., 2020) | A, T | 91.8 | 95.2 | 67.1 | 75.0 |
| MCQ (Ge et al., 2022a) | T | 89.1 | 92.3 | 65.8 | 69.8 |
| **S-ViLM** | T | **94.8** | **96.5** | **70.0** | **76.9** |

Table 4: Comparison to SOTA methods on temporal action localization (**TAL**).

| Method | TAL Task (G-TAD) | | | |
|---|---|---|---|---|
| | mAP@0.5 | @0.75 | @0.95 | Avg |
| CoCLR (Han et al., 2020) | 47.9 | 32.3 | 7.3 | 31.9 |
| XDC (Alwassel et al., 2020) | 48.4 | 32.6 | 7.6 | 32.3 |
| MoCo-v2 (Chen et al., 2020a) | 46.6 | 30.7 | 6.3 | 30.3 |
| VideoMoCo (Pan et al., 2021) | 47.8 | 32.1 | 7.0 | 31.7 |
| RSPNet (Chen et al., 2021) | 47.1 | 31.2 | 7.1 | 30.9 |
| AoT (Wei et al., 2018) | 44.1 | 28.9 | 5.9 | 28.8 |
| SpeedNet (Benaim et al., 2020) | 44.5 | 29.5 | 6.1 | 29.4 |
| PAL (Zhang et al., 2022) | 50.7 | 35.5 | 8.7 | 34.6 |
| TAC (Xu et al., 2020) | 48.5 | 32.9 | 7.2 | 32.5 |
| BSP (Xu et al., 2021c) | 50.9 | 35.6 | 8.0 | 34.8 |
| LoFi (Xu et al., 2021d) | 50.4 | 35.4 | 8.9 | 34.4 |
| TSP (Alwassel et al., 2021) | 51.3 | **37.1** | 9.3 | **35.8** |
| **S-ViLM** | **51.7** | 36.4 | **9.7** | 35.6 |

racy on UCF101 and HMDB51 is reported in Table 3. We observe that in linear probing, S-ViLM outperforms other baselines, with 3.0% and 2.9% higher than current SOTA, MMV that leverages audio and text modalities in addition on UCF101 and HMDB51. S-ViLM also achieves consistently superior performance under the fine-tuning evaluation. Outstanding performance of S-ViLM demonstrates that leveraging fine-grained video language structures during pre-training contributes to meaningful video representations. This aligns with our intuition because finer-grained video-text alignment improves video understanding.

### 4.3.4 TEMPORAL ACTION LOCALIZATION

We report the mean average precision (mAP) under different temporal Intersection over Union (tIoU) thresholds on ActivityNet in Table 4. For temporal action localization, the model is pre-trained on HowTo100M only, which is observed to be beneficial to TAL compared with VideoCC + ActivityNet (see the ablation study below). We directly use pre-trained models to extract video features as the input to G-TAD and do not further train the encoder. S-ViLM consistently exceeds other self-supervised competitors and even fully supervised approaches such as LoFi and BSP. This observation again consolidates the conclusion that vision-language pre-training can not only be applied to specific VL problems like text-video retrieval, but also benefit single-modal downstream tasks.

### 4.3.5 ABLATION STUDIES

**Pre-training datasets.** To analyze of effects of pre-training datasets, we report the model performances on selected downstream tasks in Table 5. In particular, the same model pre-trained on VideoCC achieves the best performance in zero-shot retrieval on MSR-VTT, compared with HowTo100M and WebVid-2M. These results coincide with findings in Nagrani et al. (2022), where HowTo100M has been pointed out not appropriate for vision-language tasks requiring strong alignment. S-ViLM trained on VideoCC alone significantly outperforms VCC on both tasks, showing the effectiveness of our proposed techniques. In particular, when pre-trained on the same VideoCC dataset, S-ViLM leads to better performance than MCQ. The significant improvement over MCQ shows that our techniques do help to learn better features for downstream tasks. It is also worth noting that pre-training on VideoCC and ActivityNet performs consistently better than using only one dataset, and thus we choose this setup in the main experiments.

**Training objectives.** Without loss of generality, the model in this ablation is pre-trained on VideoCC only. For better understanding S-ViLM, we start with the contrastive baseline represented in Sce-

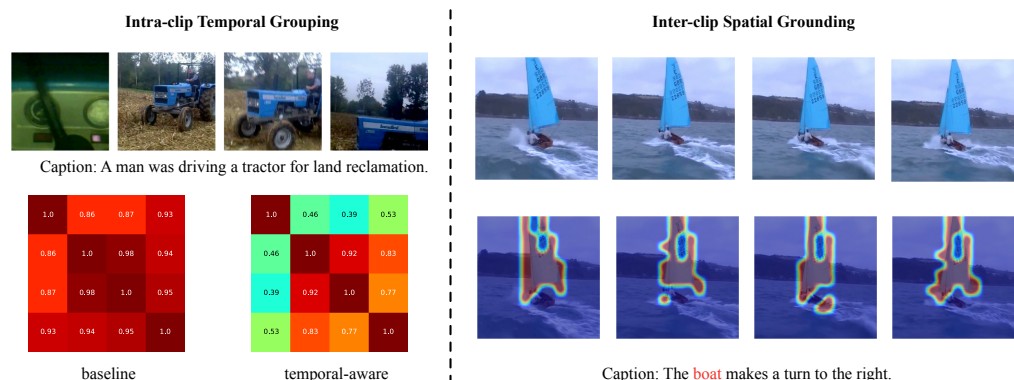

Figure 2: Visualization of S-ViLM. Left: Similarity scores of features derived from the baseline and our method. Right: Attention maps between region and object with spatial grounding.

Table 5: Effects on different choices of pre-training datasets.

| Method | PT Dataset | MSRVTT-ZS | | | TAL | | | |
|---|---|---|---|---|---|---|---|---|
| | | R@1 | R@5 | R@10 | mAP@0.5 | 0.75 | 0.95 | Avg |
| VCC | HowTo100M | 10.4 | 22.2 | 30.0 | | - | | |
| | WebVid | 15.4 | 33.6 | 44.1 | | - | | |
| | VideoCC | 18.9 | 37.5 | 47.1 | 49.9 | 34.3 | 8.7 | 33.7 |
| MCQ | VideoCC | 22.5 | 43.8 | 54.8 | | - | | |
| S-ViLM | HowTo100M | 9.4 | 22.9 | 31.3 | 51.7 | 36.4 | 9.7 | 35.6 |
| | ActivityNet | 14.4 | 33.5 | 44.0 | 50.5 | 35.3 | 8.7 | 34.5 |
| | VideoCC | 24.7 | 47.4 | 59.0 | 50.5 | 35.0 | 9.2 | 34.2 |
| | VideoCC, ActivityNet | 28.6 | 53.6 | 65.1 | 50.8 | 35.6 | 9.3 | 34.7 |

nario 1 in Table 6. Then we add our proposed spatial grouping module during the pre-training phase. This module is driven by the grouping loss $\mathcal{L}_g$ in Scenario 2, and we observe consistent improvements on all tasks across the board comparing to Scenario 1. Similarly, we introduce the temporal grouping module in $\mathcal{L}_t$ to encourage more temporal discriminative video representation. After comparing Scenario 3 to Scenario 1 in Table 6, we also observe noticeable improvements on different downstream tasks. These phenomenons suggest both spatially and temporally fine-grained features improve video understanding tasks. After combining everything together in Scenario 4, we show significant performance improvements on all tasks, which demonstrates the effectiveness of S-ViLM pre-training. Moreover, we visualize effects of temporal grouping and spatial grounding in Figure 2. It can be observed from similarity scores among frames that with temporal grouping, features from different scenes are much easier to distinguish. Besides, attention maps from spatial grounding indicates the alignment between the region and the noun phrase has been learned during the pre-training stage without any fine-grained annotations. More examples can be found in Appendix C.

Table 6: Ablation study on training objectives. We validate that our proposed spatial grounding loss and temporal grouping loss both benefit downstream tasks.

| Scenario | $\mathcal{L}_{contrast}$ | $\mathcal{L}_g$ | $\mathcal{L}_t$ | MSRVTT-ZS | | | MSVD-QA | UCF101 | TAL | | | |
|---|---|---|---|---|---|---|---|---|---|---|---|---|
| | | | | R@1 | R@5 | R@10 | Acc | Acc | mAP@0.5 | 0.75 | 0.95 | Avg |
| 1 | ✓ | | | 22.7 | 45.9 | 57.0 | 43.6 | 90.5 | 49.9 | 34.3 | 8.7 | 33.7 |
| 2 | ✓ | ✓ | | 23.3 | 46.6 | 58.6 | 44.1 | 90.6 | 50.2 | 34.7 | 8.7 | 34.0 |
| 3 | ✓ | | ✓ | 24.2 | 46.7 | 58.2 | 43.9 | 90.9 | 50.1 | 34.6 | 8.8 | 34.0 |
| 4 | ✓ | ✓ | ✓ | 24.7 | 47.4 | 59.0 | 44.9 | 91.0 | 50.5 | 35.0 | 9.2 | 34.2 |

## 5 CONCLUSION

In this paper, we present a novel video-language pre-training framework, named S-ViLM, that aims to utilize fine-grained structures in video and languages to learn region-object correspondences and temporal-aware features simultaneously. Spatial grounding and temporal grouping are introduced to achieve the goal of local region-object alignment and temporal distinction in a self-supervised manner. The proposed framework outperforms existing methods significantly on downstream tasks, including text-video retrieval, video question answering, video action recognition, and temporal action localization. The superior performance validates our design and our method could be easily scaled up, as it is self-contained and does not rely on other artifacts.

ACKNOWLEDGEMENT

We thank the reviewers for their invaluable feedbacks. Cho-Jui Hsieh is partially supported by NSF 2331966, 2325121, 2330830, 2244760, 2008173, and ONR 20230936.

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

# A   IMPLEMENTATION DETAILS

## A.1   TEXT PROMPT TEMPLATES

As mentioned in Section 3.3, we extract noun tokens by prompting noun phrases with pre-defined templates. We randomly select one from 12 templates to generate the prompt and details are presented in Table 7.

Table 7: Prompt templates used for generating noun tokens.

| Template | Prompts for noun phrases |
|---|---|
| 1 | A footage of a {}. |
| 2 | A footage of the {}. |
| 3 | A footage of one {}. |
| 4 | A video of a {}. |
| 5 | A video of the {}. |
| 6 | A video of one {}. |
| 7 | A portrait of a {}. |
| 8 | A portrait of the {}. |
| 9 | A portrait of one {}. |
| 10 | A video footage of a {}. |
| 11 | A video footage of the {}. |
| 12 | A video footage of one {}. |

## A.2   STRUCTURE OF GROUPING BLOCK

We demonstrate the structure of a grouping block in Figure 3. It features a K-means clustering attention layer, in which attention scores are computed between group tokens as query and video tokens as value. The cluster assignment is computed via gumbel softmax over group tokens and converted into a one-hot hard assignment.

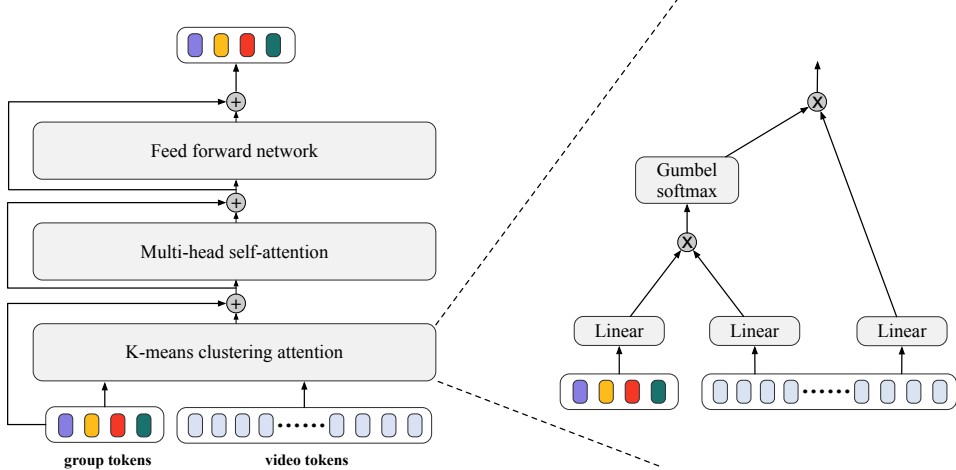

Figure 3: The structure of a grouping block. It is inserted at different layers of the video encoder to update group tokens by merging semantically similar video tokens.

## A.3   DOWNSTREAM TASKS

Implementation details of fine-tuning the pre-trained model on downstream tasks are described in this section. During fine-tuning, we resize video frames to $224 \times 224$ and sample 32 frames for each video. The maximum length for each caption is 32 by default, the same as the value in the pre-training stage. Specific optimization settings for each dataset are shown in Tables 8 to 11.

Table 8: End-to-end fine-tuning configurations on MSR-VTT for text-to-video retrieval.

| Config | MSR-VTT |
|---|---|
| optimizer | SGD |
| base learning rate | 2.5e-1 |
| optimizer momentum | 0.9 |
| learning rate schedule | cosine decay |
| batch size | 512 |
| warmup ratio | 0.1 |
| training epochs | 20 |

Table 9: End-to-end fine-tuning configurations on MSRVTT-QA and MSVD-QA for VQA.

| Config | MSRVTT-QA | MSVD-QA |
|---|---|---|
| optimizer | SGD | SGD |
| base learning rate | 1e-1 | 5e-5 |
| optimizer momentum | 0.9 | 0.9 |
| learning rate schedule | cosine decay | cosine decay |
| batch size | 64 | 64 |
| warmup ratio | 0.1 | 0.1 |
| training epochs | 30 | 30 |

Table 10: Fine-tuning configurations on UCF101 and HMDB51 for video action recognition.

| Config | UCF101 | HMDB51 |
|---|---|---|
| optimizer | SGD | SGD |
| base learning rate | 1e-1 | 1e-1 |
| optimizer momentum | 0.9 | 0.9 |
| learning rate schedule | cosine decay | cosine decay |
| batch size | 64 | 64 |
| warmup ratio | 0.1 | 0.1 |
| training epochs | 30 | 60 |

Table 11: Fine-tuning configurations on ActivityNet for temporal action localization.

| Config | ActivityNet |
|---|---|
| optimizer | Adam |
| base learning rate | 4e-3 |
| weight decay | 1e-4 |
| optimizer momentum | $\beta_1$=0.9, $\beta_2$=0.999 |
| learning rate schedule | decay by $\gamma$=0.1 every 5 epochs |
| batch size | 16 |
| training epochs | 10 |

## B    ADDITIONAL RESULTS

**Number of frames.** The number of frames used in pre-training vary among different methods. We follow the setting in Nagrani et al. (2022) to sample 32 frames for each video. Note that S-ViLM uses a vanilla vision transformer with temporal patch size 2, which effectively downsamples the video frames at the first layer. Since no downsampling happens in ALPRO and MCQ's encoder, their computational cost of 16 frames is comparable to S-ViLM with 32-frame input. In Table 12 we report 16 and 32-frame results and S-ViLM outperforms ALPRO and MCQ in both settings. As most methods use 16 frames during evaluation, we thus select to sample 32 frames to ensure a fair comparison.

Table 12: Results with different number of frames.

| Method | Input | MSRVTT-ZS | | | UCF101 | |
| --- | --- | --- | --- | --- | --- | --- |
| | | R@1 | R@5 | R@10 | Lin | FT |
| ALPRO | 8-frame | 24.1 | 44.7 | 55.4 | - | - |
| ALPRO | 16-frame | 24.7 | - | 55.0 | - | - |
| MCQ | 16-frame | - | - | - | 89.1 | 92.3 |
| S-ViLM | 16-frame | 28.5 | 53.5 | 64.0 | 93.1 | 96.0 |
| S-ViLM | 32-frame | **28.6** | **53.5** | **65.1** | **94.8** | **96.5** |

**Spatiotemporal action localization.** We report experiment results of spatial temporal action localization task on AVA v2.2 dataset. Results are presented in the table below:

Table 13: Results of spatiotemporal action localization on AVA v2.2. Two columns indicates using Detected and Grounded-truth boxes respectively.

| Method | mAP@0.5 | |
| --- | --- | --- |
| | Detected | Ground-truth |
| SlowFast | 23.80 | - |
| MViT-B | 24.50 | - |
| S-ViLM (contrastive only) | 21.95 | 26.55 |
| S-ViLM | **25.00** | **30.15** |

We can observe that S-ViLM outperforms other models and the variant with contrastive loss only when either detected boxes or ground-truth boxes are used for evaluation. This experiment demonstrates the effectiveness of the spatiotemporal learning design of our method.

## C    VISUALIZATION

We present the visualization of spatial grounding in Figure 4. For each example, we choose the group token which has the maximum similarity score of the target noun phrase, and compute the attention heatmap based on corresponding video tokens assigned to that group token. It can be observed that the alignment between the region and the noun phrase has been learned during the pre-training stage without any fine-grained annotations. In addition, more comparisons between similarity scores of baseline features and temporal-aware features are provided in Figure 5. With temporal grouping, features from different scenes are much easier to distinguish.

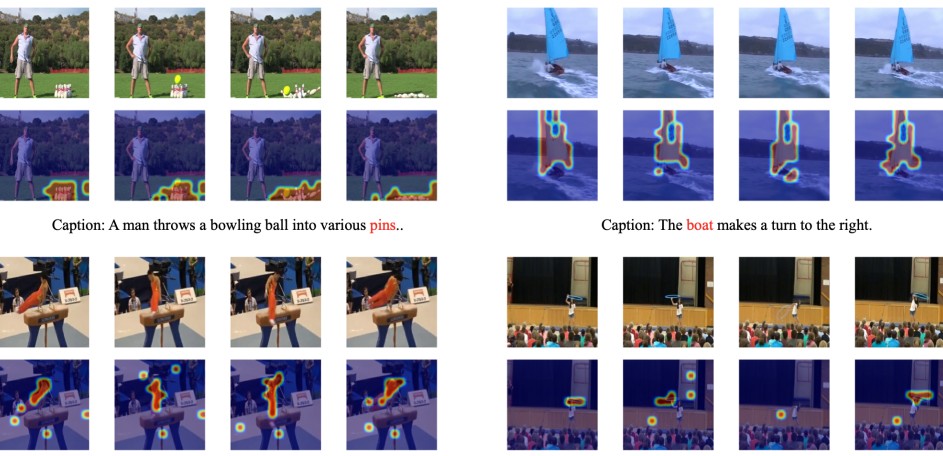

Figure 4: Visualization of spatial grounding. The attention feature map of each example is computed from the corresponding regions assigned to the group token which achieves the highest similarity score with respect to the target noun phrase.

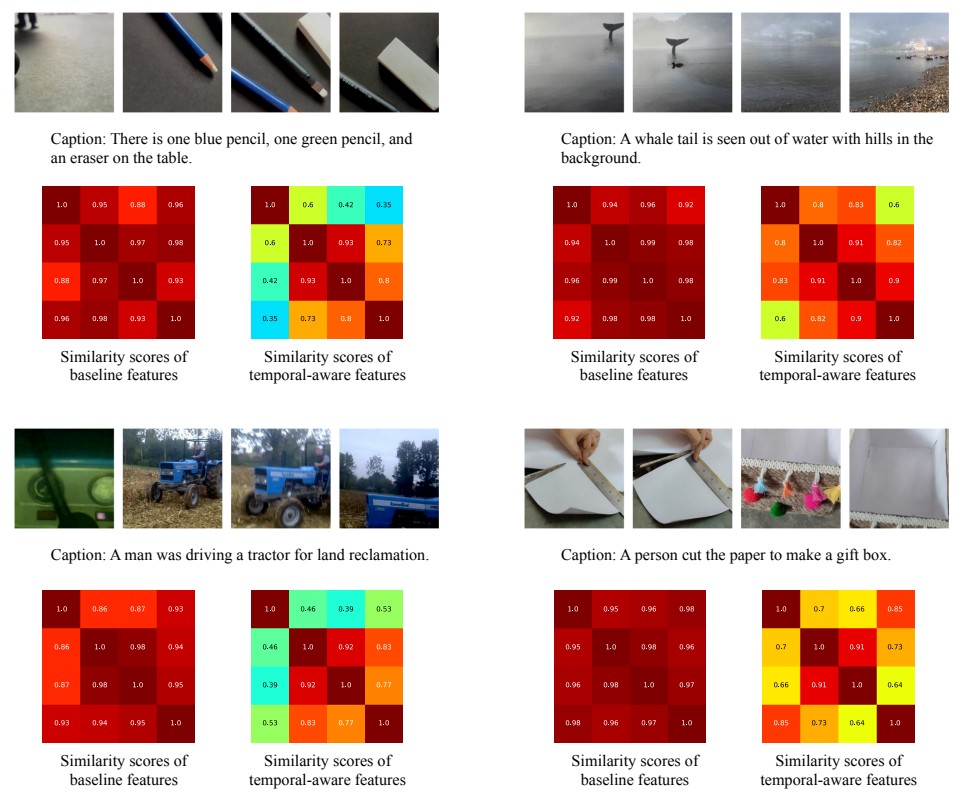

Figure 5: Visualization of temporal grouping.

