# OpenReview forum: "Structured Video-Language Modeling with Temporal Grouping and Spatial Grounding"
_ICLR.cc/2024/Conference — ICLR 2024 poster_

### Official Review · Reviewer_7b3w · 2023-10-29

**Soundness:** 3 good
**Presentation:** 3 good
**Contribution:** 2 fair
**Rating:** 6
**Confidence:** 5

**Summary:**

The paper proposes a new framework S-ViLM for video-language pre-training, which aims to improve the fine-grained understanding of both videos and text. The framework consists of two novel components: inter-clip spatial grounding and intra-clip temporal grouping. The former learns to align video regions and text objects across different video clips, while the latter learns to group video frames and text tokens within the same clip based on temporal coherence. The paper evaluates S-ViLM on four downstream tasks that require temporal localization and semantic reasoning, such as text-video retrieval, video question answering, video action recognition and temporal action localization. The paper shows that S-ViLM outperforms existing methods on these tasks and achieves state-of-the-art results. The paper also provides an extensive ablation study to demonstrate the effectiveness of each component of S-ViLM.

**Strengths:**

- The manuscript effectively conveys the concept of aligning fine-grained video and text features.
- The use of group tokens in the video encoder to align with the concepts in the text is a noteworthy approach.
- The proposed loss function significantly enhances zero-shot retrieval performance on the MSRVTT dataset.

**Weaknesses:**

- The representation of spatial concept features by the group tokens is unclear.
- The compared methods are outdated and do not compare to state-of-the-art methods.
- The claimed cut-and-paste operation has been widely used in previous studies.
- The performance of the current methods on several tasks is inferior to many recent works.

**Questions:**

- The methods being compared are outdated. It is recommended to compare them with state-of-the-art (SOTA) methods, particularly in the MSRVTT and TAL tasks. The authors can refer to https://paperswithcode.com/sota for comparing with more recent methods.
- The text may involve multiple concepts. How can we extract k noun tokens from all sentences?
- The number of group tokens remains constant during training. How do the group tokens align with different nouns in different sentences?
- The visualization results for group tokens and noun tokens should be included.
- It may be beneficial to include a grounding task since spatial alignment is claimed to be achieved.

---

> ### Author Response · Authors · 2023-11-16
> **Response to Reviewer 7b3w (1/2)**
>
> We are thankful for your valuable comments and we provide a detailed response to your questions:
>
> **Q1:** The representation of spatial concept features by the group tokens is unclear.
>
> **A1:** To make it more clear, group tokens could be regarded as a number of learnable cluster centers. These group tokens first aggregated semantically similar video tokens through our video encoder, and then were aligned with noun tokens via spatial grounding loss. Note that the ultimate group tokens to be grounded to noun tokens were conditioned on the given video, and there was no 1-1 matching between initial group tokens and noun concepts.
>
> **Q2:** The compared methods are outdated and do not compare to state-of-the-art methods. The performance of the current methods on several tasks is inferior to many recent works.
>
> **A2:** In the paper, we compared our method with VIOLETv2 [1], which was published at CVPR 2023, and it was one of the most recent works about video-language pre-training. Here we included one more recent method All-in-One [2] published at CVPR 2023 as well. We can observe in the table below that our S-ViLM still achieved competitive or even better performance in zero-shot text-to-video retrieval on MSR-VTT.
>
>
> |   Method   | Year |   R@1    |   R@5    |   R@10   |
> |:----------:|:----:|:--------:|:--------:|:--------:|
> |  VIOLETv2  | 2023 |   37.2   |   64.8   |   75.8   |
> | All-in-One | 2023 |   37.1   | **66.7** |   75.9   |
> |   S-ViLM   | 2023 | **38.4** |   65.7   | **76.3** |
>
>
> Moreover, for methods such as UCOFIA (ICCV 2023) [3] and Aurora (NeurIPS 2023) [4] they were considered as contemporaneous papers. For methods such as X-CLIP [5], and Ts2-net [6], they took advantanges of more complicated designs including much larger-scale pre-training dataset, better initialization such as CLIP-based video encoder, large language models and the like. Thus, we selected the most appropriate baselines that were reasonable and up-to-date at the same time. We also modified the part of related work to discuss those recent methods comprehensively. For TAL task, since we followed previous protocols to apply G-TAD with our frozen features, we chose the most appropriate baseline methods as well.
>
>
> **Q3:** The claimed cut-and-paste operation has been widely used in previous studies.
>
> **A3:** Although the cut-and-paste operation has been widely used in the image domain [7], it was under-explored in the video domain. To the best of our knowledge, our S-ViLM was one of the earliest attempts to leverage cut-and-paste for videos, especially for video-language pre-training. It was not trivial to adapt this operation to videos in our method, in which the cut-and-paste was not only a form of data augmentation, but also a key component in our temporal grouping loss when constructing background and foreground training examples.
>
> **Q4:** The text may involve multiple concepts. How can we extract k noun tokens from all sentences?
>
> **A4:** It should be clarified that we extracted $k$ noun phrases for each caption, instead of $k$ noun phrases for all captions in the dataset. We use spaCy to extract all noun phrases from the caption. For each caption, if the number of noun chunks are greater than $k$, we just keep $k$ of them randomly; otherwise, we choose all the nouns, and the rest of candidates are sampled from the pool with replacement.

---

> > ### Author Response · Authors · 2023-11-16
> > **Response to Reviewer 7b3w (2/2)**
> >
> > **Q5:** The number of group tokens remains constant during training. How do the group tokens align with different nouns in different sentences?
> >
> > **A5:** The correspondences between group tokens and noun tokens are learned through the spatial grounding loss in Eq.(6). Note that the number of group tokens 32 was much larger than the number of nouns within one video, and ideally different nouns with different meanings can be grounded to different group tokens for one video. Furthermore, although the number of group tokens was a constant, the ultimate group tokens to be grounded with nouns were conditioned on the video and should be distinct among different videos, and the grounding similarity was computed between the ultimate group tokens from the video $i$, $\forall i \in [1, B]$ and noun tokens from the caption $j$, $\forall j \in [1,B]$. Therefore, even with a fixed number of group tokens, it still worked excellently to learn alignment between regions and nouns in different video-caption pairs.
> >
> > **Q6:** The visualization results for group tokens and noun tokens should be included.
> >
> > **A6:** We presented some visualization results of group tokens and noun tokens in Figure 2 and Figure 4. In detail, for each example, we chose the group token which had the maximum similarity score of the target noun phrase, and computed the attention heatmap based on corresponding video tokens assigned to that group token. It can be observed that the alignment between the region and the noun phrase had been learned during the pre-training stage without any fine-grained annotations.
> >
> > **Q7:** It may be beneficial to include a grounding task since spatial alignment is claimed to be achieved.
> >
> > **A7:** Thanks for this valuable suggestion. We are running experiments on AVA v2.2, a spatiotemporal action localization task that requires spatial recognition of instances and actions, and will report results in this post and paper once ready.
> >
> > [1] Fu, Tsu-Jui, et al. "An empirical study of end-to-end video-language transformers with masked visual modeling." Proceedings of the IEEE/CVF Conference on Computer Vision and Pattern Recognition. 2023.
> >
> > [2] Wang, Jinpeng, et al. "All in one: Exploring unified video-language pre-training." Proceedings of the IEEE/CVF Conference on Computer Vision and Pattern Recognition. 2023.
> >
> > [3] Wang, Ziyang, et al. "Unified Coarse-to-Fine Alignment for Video-Text Retrieval." Proceedings of the IEEE/CVF International Conference on Computer Vision. 2023.
> >
> > [4] Wang, Haixin, et al. "Parameter-efficient Tuning of Large-scale Multimodal Foundation Model." Thirty-seventh Conference on Neural Information Processing Systems. 2023.
> >
> > [5] Ma, Yiwei, et al. "X-clip: End-to-end multi-grained contrastive learning for video-text retrieval." Proceedings of the 30th ACM International Conference on Multimedia. 2022.
> >
> > [6] Liu, Yuqi, et al. "Ts2-net: Token shift and selection transformer for text-video retrieval." European Conference on Computer Vision. Cham: Springer Nature Switzerland, 2022.
> >
> > [7] Ghiasi, Golnaz, et al. "Simple copy-paste is a strong data augmentation method for instance segmentation." Proceedings of the IEEE/CVF conference on computer vision and pattern recognition. 2021.

---

> > ### Comment · Reviewer_7b3w · 2023-11-20
> >
> > I appreciate the authors' detailed response. However, I find myself at odds with the claim that cut-and-paste operations weren't initially used in video-language pre-training. Allow me to share a few references demonstrating the implementation of cut-and-paste operations.
> >
> > [1] PAL (Unsupervised Pre-training for Temporal Action Localization Tasks, CVPR,2022).
> >
> > [2] Long-Form Video-Language Pre-Training with Multimodal Temporal Contrastive Learning. NeurIPS 2022.
> >
> > [3] Boundary-sensitive Pre-training for Temporal Localization in Videos. ICCV 2021.
> >
> > [4] Temporal perceiving video-language pre-training. Arxiv 2023
> >
> > For video-language pre-training, there is a recent method that also utilizes a similar cut-and-paste operation, which is not compared in the paper. It appears that the current method achieves lower ft (fine-tuning) but higher zs performance. I am curious to know why the performance difference between the two settings is much smaller compared to other methods.
> >
> > The Inter-clip Spatial Grounding differs from previous techniques in video-language methods, but has been explored in numerous image-language approaches. The combination of spatial grounding and the cut-and-paste operation may not seem challenging, but it still warrants further study. While the contributions may be slightly exaggerated, the exploration of the spatial-temporal dimension holds significance. However, there is currently a lack of research on spatial-temporal experiments.

---

> ### Author Response · Authors · 2023-11-21
> **Further response (1/2)**
>
> ## **Re: "cut-and-paste operation":**
> Thank you for your prompt reply with more details and references about the cut-and-paste operation. We acknowledge that they are important related works and we would like to include the following discussion in our revised manuscript. It should be clarified that _we did not claim the cut-and-paste operation as our innovation, instead our contribution is the proposed intra-clip temporal grouping loss with the cut-and-paste operation, which serves as an essential part to introduce scene changes and promote more temporal-aware feature learning._
>
> Here we would like to discuss the technique proposed in our paper and those methods you listed:
> 1. "Unsupervised Pre-training for Temporal Action Localization Tasks, CVPR,2022." _How to use synthetic videos after the cut-and-paste is different between PAL and our method._ Specifically, in PAL, clips from one video are pasted onto the other two different videos, and the goal is to align the features of pasted pseudo action regions from two synthetic videos via contrastive learning. In contrast, in our S-ViLM, we teach the model to distinguish background and foreground features within one synthetic video after cut-and-paste and formulate it as a binary classification problem for each token. Besides, compared with PAL involved in only one modality, the cut-and-paste in our method tightly integrates in video-language constrastive learning as a form of data augmentation. As shown in Eq. (6) and (7), both the grounding loss and the global contrastive loss are weighted based on the cut-and-paste operation, allowing the model to learn alignment between video and text features better.
> 2. "Long-Form Video-Language Pre-Training with Multimodal Temporal Contrastive Learning. NeurIPS 2022." _Technically, this paper does not leverage the cut-and-paste operation._ The focus of this paper is pre-training on long-form videos. The pre-traning dataset with long-form videos are constructed by consecutive clips from an existing dataset, and all videos are real ones from YouTube and there are no synthetic videos obtained from the cut-and-paste.
> 3. "Boundary-sensitive Pre-training for Temporal Localization in Videos. ICCV 2021.". _Our S-ViLM differs from this work in both the pre-training objective and strategies of the cut-and-paste operation._ For objectives, the paper designs a novel boundary-sensitive pretext (BSP) task with synthetic videos after the cut-and-paste. In particular, they construct a synthetic video by merging two videos from four types: same class, diff class, same speed and diff speed, and the training objective is to predict the merging type given the synthetic video. Instead, in our S-ViLM, we look at more fine-grained clip features and predict whether the individual clip is from the background or the foreground. As for the cut-and-paste operation, strategies in BSP depend on datasets with annotation available, while S-ViLM is fully self-supervised. In addition, as mentioned in 1., our method is about video-language pre-training and not only video representation learning.
> 4. "Temporal perceiving video-language pre-training. Arxiv 2023." _The cut-and-paste in this paper (TemPVL) applies in the feature domain instead of the raw video domain_, which means that original videos are kept intact without any modifications. They only conduct video token merging before the multi-modal encoding process, and their motivation and goal is different from ours: TemPVL tries to accurately perceive temporal boundaries in videos conditioned on the text description through a multi-modal fusion while our paper is to learn temporal-aware features via temporal grouping independent of text as shown in Figure 2 in the main paper.

---

> > ### Author Response · Authors · 2023-11-21
> > **Further response (2/2)**
> >
> > ## **Re: "questions with recent method":**
> > We assume that you are referring "recent method" to TemPVL [4]. [4] is considered as a contemporaneous work and has not been published yet. We would point out that TemPVL leverages a more complex model architecture with an additional multi-modal fusion encoder. Compared with TemPVL, our two-tower structure is advantageous in terms of training and inference efficiency.
> > As for performance difference between zero-shot and fine-tuning evaluation, we hypothesize it might come from the discrepancy of model structures. Typically, the video-language model with a fusion encoder is more expressive and it is likely that fine-tuning a model with the multi-modal fusion encoder like TemPVL would achieve better performance than a two-tower model like S-ViLM. We also note that our models are not heavily tuned due to resource constraints, which implies possibly room for improvement on fine-tuning for our method.
> >
> > ## **Re: "spatial-temporal experiments":**
> > Thanks for acknowledging our contribution of exploring the spatial-temporal fine-grained information in video-language pre-training. Per your suggestion, we include one more experiment of spatial temporal aciton localization task on AVA v2.2 dataset. Results are presented in the table below:
> >
> >
> > | Method       | mAP@0.5 (Det) | mAP@0.5 (GT) |
> > |:------------ |:-------------:|:------------:|
> > | SlowFast [1] |     23.80     |      -       |
> > | MViT-B [2]   |     24.50     |      -       |
> > | S-ViLM with contrastive loss only          |     21.95     |    26.55     |
> > | S-ViLM       |   **25.00**   |  **30.15**   |
> >
> > We can observe that our S-ViLM method outperforms baselines including the variant with contrastive loss only mAP@0.5 with both detected boxes and ground-truth boxes. This experiment demonstrates the effectiveness of spatial-temporal learning design of our method.
> >
> > [1] Feichtenhofer, Christoph, et al. "Slowfast networks for video recognition." Proceedings of the IEEE/CVF international conference on computer vision. 2019.
> >
> > [2] Fan, Haoqi, et al. "Multiscale vision transformers." Proceedings of the IEEE/CVF international conference on computer vision. 2021.

---

> > > ### Author Response · Authors · 2023-11-22
> > >
> > > Dear reviewer 7b3w,
> > >
> > > As the discussion session is approaching the end of today. We eagerly look forward to learn if we have addressed your concerns. Should you have any question, please let us know and we hope we could address it promptly.
> > >
> > > Sincerely,
> > > Authors

---

> > > > ### Comment · Reviewer_7b3w · 2023-11-23
> > > > **Comment**
> > > >
> > > > Thanks for the authors' efforts. Although authors list the differences across different methods, the introduced method shows some incremental novelty in the proposed modules. However, the spatial-temporal alignments still require further investigation. The authors provide evident results that demonstrate improvements in the generalization of the pre-training VL model. I suggest conducting more ablation studies on the spatial-temporal video experiments to validate the effectiveness of the proposed modules. Considering these points, I would like to raise my rating to borderline accept.

---

### Official Review · Reviewer_Dxbj · 2023-10-30

**Soundness:** 3 good
**Presentation:** 3 good
**Contribution:** 2 fair
**Rating:** 6
**Confidence:** 4

**Summary:**

Compared with existing video-language pre-training tasks, the authors focus on instance-level alignment with spatial and temporal granularity instead of global contrastive learning. Specifically, the authors propose two pre-training tasks, inter-clip spatial grounding and intra-clip temporal grouping, to promote learning region-object alignment and temporal-aware features simultaneously. The experiment results empirically demonstrate the effectiveness of the proposed framework.

**Strengths:**

1. The design of pre-training tasks sounds technically works. And the details of it are comprehensive.
2. The comparison and ablation study are comprehensive. The visualizations clearly present the actual contribution of inter-clip spatial grounding and intra-clip temporal grouping.
3. The overall presentation is clear and easily understandable.

**Weaknesses:**

1. The novelty may be somewhat weak.

**Questions:**

Please see the above weaknesses.

---

> ### Author Response · Authors · 2023-11-16
> **Response to Reviewer Dxbj**
>
> Thank you for your positive feedback and below we present our response to your question:
>
> **Q1:** The novelty may be somewhat weak.
>
> **A1:** Thank you for allowing us to elaborate on our work's novelty. We will explain it as follows:
>
> **a) Impact of our work:** As mentioned in Section 1, most of existing video-language pre-training frameworks focused on learning holistic representations to match a *<video, caption>* pair while fine-grained information such as region-object correspondences, or scene/action changes along the time in a video was under-explored. Nevertheless, such regional or temporal fine-grained information had been demonstrated to play a vital role in localization and reasoning tasks. Incorporting them into the pre-training paradigm was promising to further enhance the model's understanding and reasoning capabilities in both video and language domains. Thus, in our paper, we investigated how to exploit instrinsic spatiotemporally fine-grained structures contained in the pre-training dataset, and demonstrated that making use of structured video-caption interactions indeed contributed to more expressive spatiotemporal features.
>
> **b) Our technical contributions:** To leverage spatiotemporally fine-grained structures introduced above, we proposed two novel loss designs: intra-clip temporal grouping loss, and inter-clip spatial grounding loss. In particular, we adopted a cut-and-paste operation to introduce scene changes into videos. It was more frequently used in the image domain and we successfully adjusted the operation to our scenario for learning more temporal-aware features, which was a ***non-trivial*** effort. As to inter-clip spatial grounding, we proposed a soft grounding loss to capture fine-grained correspondences in a self-supervised learning, without any *<region, object>* annotations. We presented both quantitative and qualitative results in the paper to showcase the effectiveness of considering fine-grained spatiotemporal structures during the pre-training stage.

---

> > ### Author Response · Authors · 2023-11-22
> >
> > Dear reviewer Dxbj,
> >
> > As the discussion session is approaching the end of today. We eagerly look forward to learn if we have addressed your concerns. Should you have any question, please let us know and we hope we could address it promptly.
> >
> > Sincerely,
> > Authors

---

### Official Review · Reviewer_tkWv · 2023-11-01

**Soundness:** 2 fair
**Presentation:** 3 good
**Contribution:** 3 good
**Rating:** 6
**Confidence:** 5

**Summary:**

The paper proposed a new framework, namely S-ViLM, for video-language modeling.

The core ideas include:

- Perform intra-clip temporal grouping by appending clips from other videos to the start and end (cut-and-paste) and classifying each clip as foreground / background in a self-supervised manner based on feature similarity

- Perform inter-clip spatial grounding with the help of grouping blocks and tokens, based on feature similarity between prompts of nouns identified by spaCy and grouped tokens

- Joint pretraining by learning intra-clip temporal grouping,  inter-clip spatial grounding, and general video-text contrastive matching

Pretrained on VideoCC and ActivityNet-Caption, the model achieved competitive performance against previous state-of-the-arts on four downstream tasks including text-video retrieval, VQA, video action recognition and temporal action localization. Ablation studies demonstrate some important design choices and present some details on the effects of each part.

**Strengths:**

- The proposed methods are well-motivated and bring something new to video-text modeling. The intra-clip temporal grouping with cut-and-paste is an effective way of better capturing scene changes and the inter-clip spatial grounding with grouping blocks and tokens is an interesting way to enhance object (noun) understanding.

- The comparison with previous state-of-the-arts on four different downstream tasks suggest the effectiveness of the proposed method on learning more representative features.

- Ablation studies in Fig. 2 show that intra-clip temporal grouping helps the model better distinguish different scenes and inter-clip spatial grounding  leads to features that are sensitive to objects, both contributing to performance improvement as in Tab. 6.

- Good details are provided, which can make reproduction easier

**Weaknesses:**

- Extra knowledge about determining noun chunks from spaCy is introduced during pretraining, which may lead to some level of unfair comparison with other methods

- The concern regarding benefits from pretraining set are not fully addressed. Although Tab. 5 shows that S-ViLM obtains better results than VCC when both pretrained on VideoCC, it is still unclear how much benefits are brought when comparing with other methods in those downstream tasks. This may also result in unfair comparison to some extent.

- The authors claim that Eq. 3 "encourages each noun to be grounded to one or a few regions and avoids penalizing regions
that cannot find any relevant nouns." I want to hear more explanations and discussions on this. In particular, how it can prevent n_k being similar to many regions, or even all regions uniformly.

- I suggest not listing all baseline names in the main text but just citing them in the Table to save space for more important discussions and details.

**Questions:**

Please address my concerns according to the Weaknesses part.

Besides what are listed there, I also have some other questions:

- For video action recognition, I wonder if the authors have results on the Something-Something / Something-Else dataset. UCF101 and HMDB51 are both quite biased towards objects presented in the videos and therefore may benefit a lot from the inter-clip spatial grounding. I'm very curious on S-ViLM's performance on a more different action recognition dataset.

- How are start and end clip indices s, e sampled?

---

> ### Author Response · Authors · 2023-11-16
> **Response to Reviewer tkWv**
>
> Thanks for your positive and valuable comments and below we provide our response to your questions:
>
> **Q1:** Extra knowledge about determining noun chunks from spaCy is introduced during pretraining.
>
> **A1:** Noun chunks extraction primarily involves identifying and isolating the basic entities or objects within a text. It does not introduce new information but rather extracts and organizes existing information. This lightweight process is a widely-used preprocessing step in existing NLP or vision-language tasks. Specifically, noun and verb extraction have been leveraged in previous methods such as MCQ [1]. Besides, there are also some methods resorting to heavier pre-trained models in their learning processes. For example, ALPRO [2] made use of frozen CLIP encoders in their prompting entity modeling, and DemoVLP [3] utilized the off-the-shelf object detectors. Thus, we believe it is a comparatively fair comparison when only a lightweight noun extraction tool was involved in S-ViLM.
>
>
> **Q2:** The concern regarding benefits from pretraining set are not fully addressed.
>
> **A2:** Due to the restricted data access policy, the commonly-used WebVid was unavailable to us. Instead, we are running an experiment to pre-train MCQ, one of the best baselines, on VideoCC dataset. We will provide the results in the response and update the paper when the experiment is completed.
>
> **Q3:** Discussion on Eq.(3)
>
> **A3:** We want to clarify that the similarity in Eq.(3) did not prevent n_k being similar to many regions. Specifically, it was possible that a noun was grounded to multiple regions, or even as you mentioned, to all regions uniformly. In the latter extreme case, the noun might be some irrelevant object that did not appear in the video, or it was likely to occupy the whole frame, e.g., a scene full of fishes/flowers. Moreover, since there was no annotation for *<region, object>* correspondence in the pre-training dataset, we applied the soft grounding similarity shown in Eq.(3), and such a design aligned a noun with the visual contents in the video in a self-supervised manner. On the contrary, if a hard grounding was adopted here, i.e., $\max{s(\mathcal{G}, n^k)}$, the noun would be biased towards the specific region and hurt the performance significantly if such a pair was wrong. Instead, our soft version avoided penalizing regions that cannot find any relevant nouns, and always took all the regions into account. The similar idea was also leveraged in [4] about open-vocabulary image segmentation, where *<region, object>* annotations were absent as well.
>
> **Q4:** Citing baselines in the Table to save space.
>
> **A4:** Thanks for your suggestion. We have modified the paper accordingly.
>
> **Q5:** Action recognition results on Something-Something.
>
> **A5:** Thank you for your suggestion. We are running experiments on Something-Something and will update the results in the response as well as in the paper when it has finished.
>
>
> **Q6:** How are start and end clip indices s, e sampled?
>
> **A6:** For the clip sampling procedure, we first uniformly sampled the duration of the foreground video $d$ from $[N_t/2, N_t)$ to guarantee it is the majority of the blended video. Then we sampled the start index $s$ from $[0, N_t-d)$, and the end index $e$ was computed naturally as $e = s + d$. We included this detail in our latest version.
>
> [1] Ge, Yuying, et al. "Bridging video-text retrieval with multiple choice questions." Proceedings of the IEEE/CVF Conference on Computer Vision and Pattern Recognition. 2022.
>
> [2] Li, Dongxu, et al. "Align and prompt: Video-and-language pre-training with entity prompts." Proceedings of the IEEE/CVF Conference on Computer Vision and Pattern Recognition. 2022.
>
> [3] Cai, Guanyu, et al. "Revitalize Region Feature for Democratizing Video-Language Pre-training of Retrieval." arXiv preprint arXiv:2203.07720 (2022).
>
> [4] Ghiasi, Golnaz, et al. "Scaling open-vocabulary image segmentation with image-level labels." European Conference on Computer Vision. Cham: Springer Nature Switzerland, 2022.

---

> > ### Author Response · Authors · 2023-11-21
> > **Additional results to Reviewer tkWv**
> >
> > Dear Reviewer tkWv,
> >
> > Thanks for your patience. As promised in the previous response, we now present additional results to support the effectiveness of our method.
> > * For Q2 about the pre-training dataset, we ran an experiment to pre-train the MCQ baseline on VideoCC dataset. Zero-shot text-video retrieval results on MSR-VTT are reported below:
> >
> > | Method |   PT Dataset    | R@1  | R@5  | R@10 |
> > | ------ |:---------------:|:----:|:----:|:----:|
> > | MCQ    |     VideoCC     | 22.5 | 43.8 | 54.8 |
> > | S-ViLM |     VideoCC     | **24.7** | **47.4** | **59.0** |
> >
> > It can be observed that when pre-trained on the same dataset, S-ViLM still leads to better performance than MCQ. The significant improvement over MCQ shows that our techniques, temporal grouping and spatial grounding, do help to learn better features for downstream tasks.
> >
> > * For Q5 about action recognition on Something-Something, we fully fine-tune the pre-trained model on the Something-Something V2 dataset and the detailed results are presented below:
> >
> >
> > | Method          | Top-1 Acc | Top-5 Acc |
> > |:--------------- |:---------:|:---------:|
> > | TimeSformer [1] |   59.5    |     -     |
> > | Frozen [2]      |   61.6    |   85.7    |
> > | S-ViLM with contrastive loss only             |   60.2    |   84.8    |
> > | S-ViLM          | **63.5**  | **87.2**  |
> >
> > Compared with baseline methods, S-ViLM performs consistently better on the Something-Something V2 dataset for action recognition.
> >
> > [1] Bertasius, Gedas, Heng Wang, and Lorenzo Torresani. "Is space-time attention all you need for video understanding?." ICML. Vol. 2. No. 3. 2021.
> >
> > [2] Bain, Max, et al. "Frozen in time: A joint video and image encoder for end-to-end retrieval." Proceedings of the IEEE/CVF International Conference on Computer Vision. 2021.

---

> > > ### Author Response · Authors · 2023-11-22
> > >
> > > Dear reviewer tkWv,
> > >
> > > As the discussion session is approaching the end of today. We eagerly look forward to learn if we have addressed your concerns. Should you have any question, please let us know and we hope we could address it promptly.
> > >
> > > Sincerely,
> > > Authors

---

> > > > ### Comment · Reviewer_tkWv · 2023-11-22
> > > >
> > > > I appreciate the authors' efforts on the rebuttal. It addressed most of my concerns.
> > > >
> > > > After reading other reviews, I agree that the novelty is not particularly high, yet I feel the good performance across various downstream tasks and on multiple different datasets show the effectiveness of the attempts, yielding some merits.
> > > >
> > > > I will keep my rating unchanged.

---

### Official Review · Reviewer_adH9 · 2023-11-01

**Soundness:** 3 good
**Presentation:** 2 fair
**Contribution:** 3 good
**Rating:** 8
**Confidence:** 4

**Summary:**

The paper addresses improving video-language alignment. The paper introduces two novel pre-training tasks, namely, inter-clip spatial grounding and intra-clip temporal grouping. The inter-clip spatial grounding aims to associate relevant image regions with text. This task is weakly supervised, without explicit correspondences between regions and nouns. Instead, it employs learnable 'group tokens' to cluster semantically similar image regions.The intra-clip temporal grouping optimizes the model features to be able to distinguish a video clip (start/end time) from a background clip, akin to a metric learning loss approach. The benefit of these tasks is that the supervision can be generated automatically using random cut & paste and pre-processing the captions. The experimental results show that the proposed method outperforms VCC (Nagrani et al., 2022) on multiple benchmarks. The ablation studies on the two introduced losses indicate that each contributes to improving the original method. However, when combined, their combined effect results in a marginal improvement compared to employing each loss separately.

**Strengths:**

S1. The proposed additional pre-training tasks (inter-clip spatial grounding and intra-clip temporal grouping) are reasonable. The supervision can be generated automatically using random cut & paste and pre-processing the captions for extracting the nouns.

S2. I liked that the authors provide Table 5 that shows the effects on different choices of pre-training datasets. This allows comparing the proposed method and VCC trained on the same dataset (HowTo100M and VideoCC). It is good to know that the proposed method is on par with VCC when trained on HowTo100M, but significantly outperforms VCC when trained on VideoCC on MSRVTT-ZS.

S3. The authors provide an ablation study on combinations of their proposed losses.

S4. The paper presents visualizations of the affinity scores of learned features, alongside the attention map of S-ViLM in Figure 2. These visualizations are interesting and further validate that the model was trained as intended.

**Weaknesses:**

W1. The authors state that most video-language pre-training methods neglect scene/action changes along the time in a video. However, there are works like LaViLa [a] that takes temporal and patch-level information into account.
[a] Learning Video Representations from Large Language Models, Zhao et al., CVPR 2023

W2. Some design choices are not obvious but no ablation study was presented in the paper.
(1) Instead of Eq 1-2, I am curious if the authors tried BCE loss on z_i clip using the mask m_i. If so, how did the performance differ?
(2) Instead of using grouping blocks, one can opt for existing region proposal networks or segmentation networks to obtain semantically similar regions, and then aggregate visual features within those regions to compute relevance against the nouns.

W3. The writing could be further improved for enhanced clarity

- It would be nice if "interaction" is clarified in “c) modality fusion and interaction” in the second paragraph of Section 1.
- The term "region-object groundingness" is ambiguous in “Specifically, group tokens aggregate semantically similar video tokens via grouping blocks to promote region-object groundingness and video tokens are utilized in temporal grouping to improve temporal awareness.”
- I suggest using “semantically similar regions” in “Thus, we adopt M learnable group tokens to cluster semantic similar regions in a self-supervised manner.”

**Questions:**

Q1. See W2 (1).
Q2. In Eq 1, were z_i’s L2 normalized?

---

> ### Author Response · Authors · 2023-11-16
> **Response to Reviewer adH9**
>
> We appreciate your constructive reviews for our paper. Here are our detailed responses to your questions:
>
> **Q1:** There are works like LAVILA that takes temporal and patch-level information into account.
>
> **A1:** Thanks for pointing out the relevant method. The focus of LAVILA was to generate more diverse and dense captions for video dataset through automatic annotation from large language models. Compared with LAVILA which incorporated temporal information implicitly, our S-ViLM focused on pretraining method designs, with novel losses, intra-clip temporal grouping and temporal cut-and-paste augmentation, to improve video representation learning. We expect that integrating these two directions can lead to further improvement and we have added a discussion of LAVILA in **Related Work** in the modified paper.
>
> **Q2:** Some design choices are not obvious but no ablation study was presented in the paper.
>
> **A2: (1)** For Eq. (2), we tried BCE loss in our preliminary experiments and it achieved similar performance to that of MSE loss presented in the paper. This was an implementation choice on how to deal with the binary classification of foreground and background clips. Since both MSE and BCE worked well in distingushing foreground and background clips and no obvious difference was observed, we selected a relatively simple MSE loss for temporal grouping. In Eq. (1), $z_i^\text{clip}$, $z_i^b$ and $z_i^f$ were $L_2$ normlized, and it was actually cosine similarity instead of pure dot product following previous protocols. Softmax with a temperature factor $\tau=0.05$ was applied. We highlighted this correction in blue in the paper.
> **(2)** The idea to leverage existing region proposal networks has been implemented in DemoVLP [1], and we compared it with our S-ViLM in Table 1. It can be observed that our method significantly outperforms DemoVLP for text-video retrieval on MSR-VTT. On the other hand, using off-the-shelf region proposal networks or segmentation networks may not be a good choice due to following reasons: **a)** applying the model to a large-scale video dataset was time-consuming; **b)** the model was trained on the dataset from different domains and might generalize poorly on video datasets; **c)** we cannot further train the model for adaptation to our data because of lack of region annotations in the pre-training dataset. Instead, adopting grouping blocks to aggregate semantically similar regions in a _self-supervised manner_ contributed to more dynamic and flexible learning, and was a more suitable and efficient choice in video-language pre-training without region annotations.
>
> **Q3:** The writing could be further improved for enhanced clarity.
>
> **A3:**
> * Sorry for the confusion about "interaction" here. We want to clarify that modality fusion and interaction referred to the same concept that different modalities were further encoded together rather than separately as in the step a). In particular, we can either use cross-attention or feed a long sequence by concatanating different modalities into an additional encoder. In our paper, we skipped this step and simply projected representations from video and language into the same latent space for the subsequent pre-training.
> * **Region-object groundingness** indicates the alignment between a region in the video and an object described in the caption. Specifically, as illustrated in inter-clip spatial grounding in Figure 1, the word “pins” in red corresponds to the red region in the left frame. The sentence should be rephrased as: _"Group tokens aggregate semantically similar video tokens via grouping blocks and are then aligned with object concepts by spatial grounding. It promotes region-object groundingness, which indicates the alignment between a region in the video and an object in the caption, e.g., as illustrated in Inter-clip Spatial Grounding in Figure 1, the red region corresponds exactly to the word “pins” in red."_
> * Thanks for your suggestion. We have corrected the grammar error in our latest version.
>
> [1] Cai, Guanyu, et al. "Revitalize Region Feature for Democratizing Video-Language Pre-training of Retrieval." arXiv preprint arXiv:2203.07720 (2022).

---

> > ### Comment · Reviewer_adH9 · 2023-11-22
> >
> > Thanks for clarification. I think it would be helpful to also including the reasoning about the about design choices in the paper as well (Q2).

---

> > > ### Author Response · Authors · 2023-11-22
> > >
> > > Thanks for your suggestion. We have included the reasoning to corresponding parts in the latest version.

---

### Author Response · Authors · 2023-11-22
**We look forward for your feedback!**

Dear Reviewers,

As the reviewer-AC discussion phase will end soon, we cordially seek feedback on our response to your reviews. Please let us know if there are outstanding issues, and we are eager to address them promptly. Your constructive input is highly valued, and we appreciate your commitment to improving our manuscript.

Thank you for your time and consideration, Authors

---

### Meta-Review · Area_Chair_ErpF · 2023-12-09

**Metareview:**

Paper proposes an approach for improved video-language alignment. In doing so it introduces two novel pre-training tasks which involve spatial and temporal grounding. Paper was reviewed by four reviewers and received the following ratings: 1 x Accept, 3 x Marginally above Accept. Generally, all reviewers agree that the proposed approach is valuable and performs well in practice. Most of the technical concerns and requested additional comparisons (including results on Something-Something dataset) have been addressed in the rebuttal. However, reviewers remain somewhat concerned about the relatively incremental novelty.

AC has read the reviews, rebuttal and discussion that followed, as well as looked through the paper itself. AC agrees with the reviewers that while approach performs well in practice and formulation is very sensible, the overall novelty is somewhat limited. Nether-the-less, given uniformly positive reviews, on balance, AC believes that the work would still be useful for the community. Therefore the decision is to accept the paper as a Poster.

**Justification For Why Not Higher Score:**

Technical novelty is somewhat limited.

**Justification For Why Not Lower Score:**

The approach is very sensible and results are state-of-the-art. The experiments are also quite thorough.

---

### Decision · Program_Chairs · 2024-01-16

Accept (poster)